behaviour, ecology

animal distributions, habitat use, biologging, ticks, trade-off, normalized difference vegetation index

**Author for correspondence:**
Caroline Liddell
e-mail: caroline.liddell@bristol.ac.uk

# Response to resources and parasites depends on health status in extensively grazed sheep

Caroline Liddell[1], Eric R. Morgan[1,2], Katie Bull[3] and Christos C. Ioannou[1]

[1]School of Biological Sciences, University of Bristol, Bristol BS8 1TQ, UK
[2]School of Biological Sciences, Queen's University Belfast, Belfast BT9 5BL, UK
[3]Bristol Veterinary School, University of Bristol, Bristol BS40 5DU, UK

CL, 0000-0002-7643-5356; ERM, 0000-0002-5999-7728; CCI, 0000-0002-9739-889X

A fundamental question in animal ecology is how an individual's internal state and the external environment together shape species distributions across habitats. The increasing availability of biologgers is driving a revolution in answering this question in a wide range of species. In this study, the position of sheep (*Ovis aries*) from Global Positioning System collars was integrated with remote sensing data, field sampling of parasite distributions, and parasite load and health measures for each tagged individual. This allowed inter-individual variation in habitat use to be examined. Once controlling for a positive relationship between vegetation productivity and tick abundance, healthier individuals spent more of their time at sites with higher vegetation productivity, while less healthy individuals showed a stronger (negative) response to tick abundance. These trends are likely to represent a trade-off in foraging decisions that vary between individuals based on their health status. Given the rarity of studies that explore how animal distributions are affected by health and external factors, we demonstrate the value of integrating biologging technology with remote sensing data, traditional ecological sampling and individual measures of animal health. Our study, using extensively grazed sheep as a model system, opens new possibilities to study free-living grazing systems.

## 1. Background

Animals are distributed according to spatial variation in resources, such as forage [1,2], and the suitability of environmental parameters [3]. Their distribution can be further altered by other external factors such as predation and parasitism [4,5]. Advances in animal-mounted sensors such as Global Positioning System (GPS) biologgers are allowing fundamental ecological questions on how species are distributed to be answered in a greater range of species and environments than has been previously possible [6]. As GPS provides spatial location across time for tagged individuals, such data can be integrated with traditional ecological sampling methods (e.g. transect or quadrat sampling), data from remote sensing (e.g. topographic surveys and estimating above ground biomass) and characteristics of the animals that have been tagged (e.g. sex, age and breeding status). Together, this can reveal species preferences and explain distributions of animals at a finer scale than traditional habitat distribution maps and allows for greater power in models predicting range shifts in response to environmental change. However, this approach has yet to consider the internal state variables of health and parasite load of individuals in determining their distribution within landscapes.

Parasites are an important yet often overlooked ecological component of natural as well as agricultural systems [7,8]. While parasites do not usually cause the same level of mortality as predators, they do negatively affect the

survival and reproductive rates of their host, and interactions with parasites occur more frequently than predation events [9–11]. Any behavioural mechanism of the host in response to parasites that reduces the risk of encounter would be advantageous to the host's fitness [8,12–17]. Because the distribution of parasites in space and time influences their likelihood of being encountered, host behavioural defences will depend on the predictability of the parasites' locations. This is particularly true because many parasites of terrestrial animals have lower mobility than their hosts, making their successful transmission highly dependent on the movement of the host rather than on the movement of the parasites themselves [13,16,18]. In general, parasites such as ticks and gastrointestinal nematodes are dependent on specific abiotic and biotic conditions, and therefore are associated with specific habitat types or environmental features that enhance their development and survival, as well as on the past distribution of infected hosts.

A number of tick species are distributed throughout the UK and northern Europe, and are increasing in abundance, expanding their ranges and carrying a growing list of pathogens [19,20]. The most common tick is *Ixodes ricinus*, which has a complex life cycle including multiple hosts and free-living stages [21], and preferentially inhabits dense matted vegetation in which humidity and warmth are sufficient and relatively stable [19,21]. To quest for hosts, ticks climb up vegetation and attach to an animal as it passes by. Hosts may be able to avoid taller, denser vegetation where ticks are typically found to reduce encounter rates and hence infection risk. Alternatively, they may avoid ticks directly, as has been shown in cattle, which avoid grazing in paddocks with high densities of tick larvae [22], as well as deer [23] and some smaller mammals (squirrels and racoons), which give up larger amounts of food from experimental feeding trays at sites with higher tick abundances [11].

Given spatial variation in parasite risk and the quantity and quality of forage, herbivores face complex choices when trading off maximizing food intake versus minimizing infection risk. The relationship between parasite distribution and foraging behaviour is potentially made more complex by the health or parasitic infection status of individual hosts [9]. Indeed, in their study into the influence of parasite burdens on movement, Falzon *et al.* [24] found a significant positive correlation between individual parasite burden (measured as nematode faecal egg count, FEC) and the overall distance travelled by GPS-collared sheep over a 24 h period. More recently, Högberg *et al.* [25] recorded an increased number of lying bouts in cattle suffering from high experimentally induced gastrointestinal nematode infections, compared with dewormed cattle. Altered use of space by infected hosts then has the potential to feed forward to subsequent parasite distribution in the environment. Thus, while animals clearly respond to the distribution of resources and risks in their habitat, it is essential to establish how health status may modify these responses.

In this study, we use sheep (*Ovis aries*) extensively grazing in an upland moorland as a model system. We explored the factors influencing the spatial distribution of individuals tagged with GPS collars by combining location data with individual-level measures of health and parasite load, remote sensing of terrain steepness and primary productivity (as measured by the normalized difference vegetation index, NDVI), and ecological sampling of tick distributions across the habitat. The use of GPS allows high-resolution spatial and temporal tracking of individuals over extended periods of time. Hence, the foraging decisions of individuals can be inferred from their spatial distribution as a function of internal and external biological variables and physical features of the habitat. Specifically, we aimed to determine how spatial distribution is affected by the presence of ticks and the health and internal parasite status of individuals.

## 2. Methods

### (a) Study area

The study area is located at 50°29′16.5″ N, 3°59′16.7″ W, near Sheepstor in the southwest of Dartmoor National Park in Devon, UK. The area is approximately 5 km², loosely bounded by a river in the south and east, and by small roads in the north and west. It ranges in elevation from 217 to 476 m.a.s.l. The vegetation consists mainly of grassland and bracken. In the summer months, when this study was carried out, livestock over the whole of the Sheepstor area comprised approximately 470 ewes plus lambs (*O. aries*), 75 cows (*Bos taurus*), most of which had calves, as well as some ponies (*Equus caballus*) occasionally straying into the area.

### (b) Procedure

Sheep were herded from the moor and returned to a holding pen on the farm in Sheepstor on the first morning of the study (14 June 2017). Out of this flock of 80 Scottish blackface ewes with 20 ewe lamb replacements and almost 120 suckling lambs, a total of 30 ewes were tagged with GPS trackers (igot-U GT-600, MobileAction Taiwan) attached to custom-made Velcro collars. Sheep were tagged in the order of their appearance in a race after being mixed in the holding pen. The GPS trackers were set to record their location every 2 min. At this recording interval, the battery life of the trackers was two weeks. The tracker collars were removed 12 days later, on 26 June 2017, but data were excluded after 22 June due to the sheep being moved back to the farm on that date. Recordings from the day of attachment (14 June 2017) were also discarded as sheep were returned to the grazing area on this day, which would affect their location in the grazing area. Data were downloaded using @trip software (MobileAction Technology 2018, v. V5.0.1601.472) and exported as .csv files to be processed in R (R Core Team 2017, v. 3.4.0) and QGIS (QGIS Development Team 2017, v. 2.18.9).

Of the 30 sheep tagged with GPS trackers, data from seven were either lost or discarded due to lack of sufficient recordings. To correct for errors in location recordings by the GPS trackers, 281 locations recorded outside of the grazing area were discarded. This left a total of 114 093 location recordings of 23 sheep for further analysis.

### (c) Parasite load and health

During initial collar attachment, faecal samples from each tagged sheep were collected. Within 2 days, the number of gastrointestinal nematode eggs per gram was determined using the mini-FLOTAC method [26] and expressed as FEC in eggs per gram (epg). Sheep were also assigned FAMACHA (Faffa Malan Chart) scores based on the colour of the conjunctival mucous membrane, which indicates the degree of anaemia and can be influenced by severity of parasitic nematode infection [27]. However, heavy tick infestations can also cause significant blood loss [21]. Therefore, we used FAMACHA as a general indicator of health rather than as a measure of nematode infection, and FAMACHA and FEC may not be correlated if anaemia is not being primarily caused by nematode infection. All procedures

were approved by the University of Bristol Ethical Review Group (UIN UB/16/076).

## (d) Tick sampling

A tick abundance survey was carried out on 24 July 2017 at 25 sample sites across the study area. Sampling sites were determined by creating a grid of evenly spaced points across the study area in QGIS. A 1 m$^2$ white cotton blanket was dragged across ground vegetation for 10 m over three parallel transects per sampling site. Each transect was 10 m apart. The number of ticks of each stage (larvae, nymphs and adults) collected on the blanket was recorded. As the vast majority of the recorded ticks (99%, $n = 712/719$) were larvae, the numbers of ticks in all stages for the three transects were summed to a single value per sampling site.

## (e) Satellite data

Satellite data were obtained from Sentinel-2, a polar-orbiting, multispectral high-resolution imaging mission for land monitoring developed by the European Space Agency (ESA) as part of the Copernicus Programme. The satellite Sentinel-2A collects data in 13 spectral bands with resolutions of 10–60 m in a 290 km swath and with high revisit frequency [28]. The data collected by this satellite are freely available to download from the Copernicus open access hub (https://scihub.copernicus.eu/dhus/#/home). The Sentinel-2A dataset used in this study was sensed on 25 May 2017 and was selected because it is a cloud-free image taken within a month of obtaining sheep GPS locations.

## (f) Normalized difference vegetation index and terrain steepness

NDVI is used as a measure of primary productivity of vegetation. It is calculated as (NIR − Red)/(NIR + Red), where NIR is the near-infrared band and Red is the infrared band. Green vegetation strongly absorbs light in the red spectral region while reflecting light in the near-infrared region, resulting in high values of NDVI [29]. NDVI has been shown to be strongly correlated with photosynthetic capacity and productivity of vegetation and therefore is a good measure of forage availability for herbivores [1,29]. The spectral bands (band 8 and band 4, resolution = 10 m) used for these calculations were sensed by Sentinel-2A and processed in QGIS 2.18.9. NDVI was exported from QGIS as a TIF file for further processing in R. Terrain steepness was calculated from Digital Elevation Models (DEM) in QGIS.

## (g) Statistical analysis

As the tick samples were effectively samples at single locations, we created a buffer zone around each of these [30]. The buffer zone around each tick sampling site was used as the spatial unit of analysis for the tests, and buffer zone diameters of 30, 50, 75 and 100 m were tested to explore whether any of the effects were sensitive to spatial scale [30–32].

All data analyses were carried out in R v. 3.5.1. To determine the importance of external variables such as terrain steepness, forage quality (as measured by NDVI) and tick abundance on the spatial distribution of sheep, and whether internal variables such as FAMACHA and FEC affected the response of sheep to these variables, generalized linear mixed models (GLMMs) were used with the number of sheep recordings in each buffer zone as the response variable, and sheep ID included as a random intercept. Using the glmmTMB function from the glmmTMB package, we fitted GLMMs and zero-inflated GLMMs to the sheep recording data with Poisson and negative

binomial distributions on the main-effects-only model. Models were compared with one another using the difference in the corrected Akaike information criterion (ΔAICc) using the AICtab function from the bbmle package. We found the zero-inflated negative binomial model which takes into account the data containing more zeros than expected from typical error distributions [33] had the best fit and was therefore applied to the models used in further analysis (table 1). These included a null model with no explanatory variables, the main-effects-only model (with all five explanatory variables) and six models including all main effects and each with one of the six possible interaction terms between the external variables (terrain steepness, NDVI and tick abundance) and health variables (FAMACHA and FEC). Including these interactions (for example, the terrain steepness × FAMACHA interaction) explicitly tests whether individuals' responses to the external variable were dependent on their health status. The lower a model's AIC value, the more likely the model explains the variation in the data, with the most likely model having a ΔAICc value of zero. There is substantial support for models with ΔAICc values within two units of the most likely model [34], thus any interaction term in models with an AICc two units lower than the main-effects only model indicate that the addition of the interaction term explains an important proportion of the variance in the data.

# 3. Results

## (a) Internal and external variables

For the 23 sheep, the mean FEC was 66.5 epg (range 0–215) and the mean FAMACHA score was 2 (range 1–3, on a 5-point scale with half scores (e.g. 2.5) possible. Lower scores indicate less anaemia and better health). A total of 719 *I. ricinus* ticks were collected across the 25 sample sites. The average number of ticks per site was 28.8 (equivalent to a density of *ca* 1 m$^{-2}$ as an area of 30 m$^2$ was dragged per site), ranging from 1 to 295 (0.03 to 9.83 m$^{-2}$). NDVI ranged from 0.489 to 0.678 with higher values indicating greener vegetation. Terrain steepness ranged from 2.352 to 13.263, with higher values indicating steeper terrain.

There was a significant positive relationship between tick abundance and NDVI in buffer zones of 30 and 100 m (Spearman's $\rho = 0.402$, $p = 0.047$ and $\rho = 0.414$, $p = 0.040$, $n = 25$, respectively; figure 1) and a close to significant correlation in buffer zones of 50 and 75 m (Spearman's $\rho = 0.388$, $p = 0.056$ and $\rho = 0.393$, $p = 0.052$, $n = 25$, respectively). There was no relationship between tick abundance and mean steepness of the terrain in buffer zones of any size ($\rho < 0.2$, $p > 0.3$, $n = 25$). However, there was a significant positive relationship between NDVI and mean steepness of the terrain in buffer zones of all sizes except the smallest (Spearman's $\rho = 0.435$, 0.435 and 0.454, $p = 0.031$, 0.031 and 0.024 (50, 75 and 100 m); for 30 m, $\rho = 0.381$, $p = 0.061$ (30 m), $n = 25$). There was no relationship between FEC and FAMACHA (Spearman's $\rho = 0.312$, $p > 0.1$, $n = 27$).

## (b) Distribution of sheep

The existence of relationships between the three external variables (NDVI, terrain steepness and tick abundance) justifies including all main effects in the models as this allows us to test, for example, whether the spatial distribution of sheep is associated with the abundance of ticks after controlling for NDVI (which correlates positively with tick abundance). In general, the AICc model comparisons

**Table 1.** Comparison of the models explaining the variance in spatial distribution of sheep across their grazing landscape. Each model differs based on the explanatory variables included. The main-effects only model (m Main) includes all the main explanatory variables only, namely NDVI, tick abundance, steepness, FEC and FAMACHA. The model m Null contains no explanatory variables. All the other models include the main effects plus an additional interaction term after which they are named, e.g. m NDVI × FAMACHA includes all the main explanatory variables as well as the interaction between NDVI and FAMACHA. The ΔAICc refers to the difference in the corrected Akaike information criterion between the model and the most likely model which has a ΔAICc value of zero. d.f. refers to degrees of freedom. Models are ordered by increasing ΔAICc. Likely models (i.e. those with ΔAICc values within two units of zero) have been highlighted in italics. Results are shown for models using data at the four different spatial scales (buffer zone diameters around each tick sampling point).

| model | ΔAICc | d.f. | model | ΔAICc | d.f. |
|---|---|---|---|---|---|
| **30 m** | | | **50 m** | | |
| *m NDVI × FAMACHA* | *0.0* | *10* | *m NDVI × FAMACHA* | *0.0* | *10* |
| m NDVI × FEC | 11.6 | 10 | m Ticks × FAMACHA | 10.1 | 10 |
| m Ticks × FAMACHA | 12.4 | 10 | m Main | 13.9 | 9 |
| m Main | 14.4 | 9 | m Steepness × FAMACHA | 14.0 | 10 |
| m Steepness × FAMACHA | 14.6 | 10 | m NDVI × FEC | 14.1 | 10 |
| m Ticks × FEC | 15.6 | 10 | m Steepness × FEC | 15.2 | 10 |
| m Steepness × FEC | 15.8 | 10 | m Ticks × FEC | 15.6 | 10 |
| m Null | n.a. | 4 | m Null | 104.5 | 4 |
| **75 m** | | | **100 m** | | |
| *m NDVI × FAMACHA* | *0.0* | *10* | *m NDVI × FAMACHA* | *0.0* | *10* |
| m Ticks × FAMACHA | 2.2 | 10 | *m Steepness × FAMACHA* | *0.0* | *10* |
| m NDVI × FEC | 3.2 | 10 | *m Main* | *1.4* | *9* |
| m Main | 4.1 | 9 | *m Ticks × FAMACHA* | *2.0* | *10* |
| m Steepness × FAMACHA | 4.7 | 10 | m Ticks × FEC | 2.5 | 10 |
| m Ticks × FEC | 5.8 | 10 | m Steepness × FEC | 3.0 | 10 |
| m Steepness × FEC | 5.8 | 10 | m NDVI × FEC | 3.1 | 10 |
| m Null | 116.7 | 4 | m Null | 120.6 | 4 |

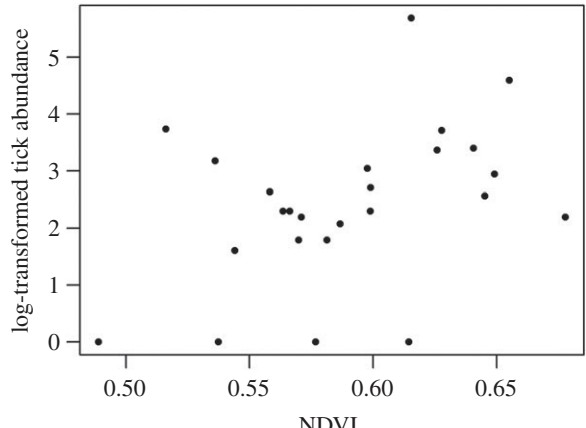

**Figure 1.** Relationship between tick abundance and NDVI in buffer zones of 50 m around the sampling point.

revealed that sheep distributions depended on NDVI, but that this varied with the health of each individual (i.e. models with the interaction term NDVI × FAMACHA were the most likely model at all buffer zone sizes; table 1). Plotting the effect of NDVI on sheep locations reveals how this relationship depends on the health status (FAMACHA score) of the sheep (figure 2). Sheep were more often recorded in areas with higher mean NDVI scores, as the effect of NDVI was always positive. However, the locations of healthier sheep (i.e. those with lower FAMACHA scores) were more strongly associated with NDVI than less healthy sheep (figure 2, top row), implying that healthier sheep were more sensitive to the distribution of vegetation quality than less healthy individuals.

Health status, as measured by the FAMACHA score, was also important in how sheep responded to tick abundance across the study area. The model with the tick abundance × FAMACHA interaction was more likely than the main-effects-only model in buffer zones of 30, 50 and 75 m (with a ΔAICc approx. or greater than two compared to the main effects model) and within two units of the most likely model in the 100 m buffer zone size. In contrast with the effect of NDVI, sheep distribution was negatively associated with tick abundance, so that sheep were less likely to be found in areas with more ticks. However, this effect was stronger in less healthy sheep (i.e. those with higher FAMACHA scores), indicating a stronger avoidance of tick-infested locations in less healthy individuals (figure 2, bottom row).

Across the different buffer zone sizes, these results imply that sheep health as measured by the internal health variable FAMACHA had an impact on variation in sheep spatial distribution in response to the external variables NDVI and tick abundance. At a buffer zone of 100 m, the main-effects-only model was within two AICc units of the most likely model, implying weaker support for any of the models with interaction terms than in the smaller buffer zone datasets. No model with an FEC interaction term was consistently more likely than the main-effects-only model in more than two buffer zone sizes.

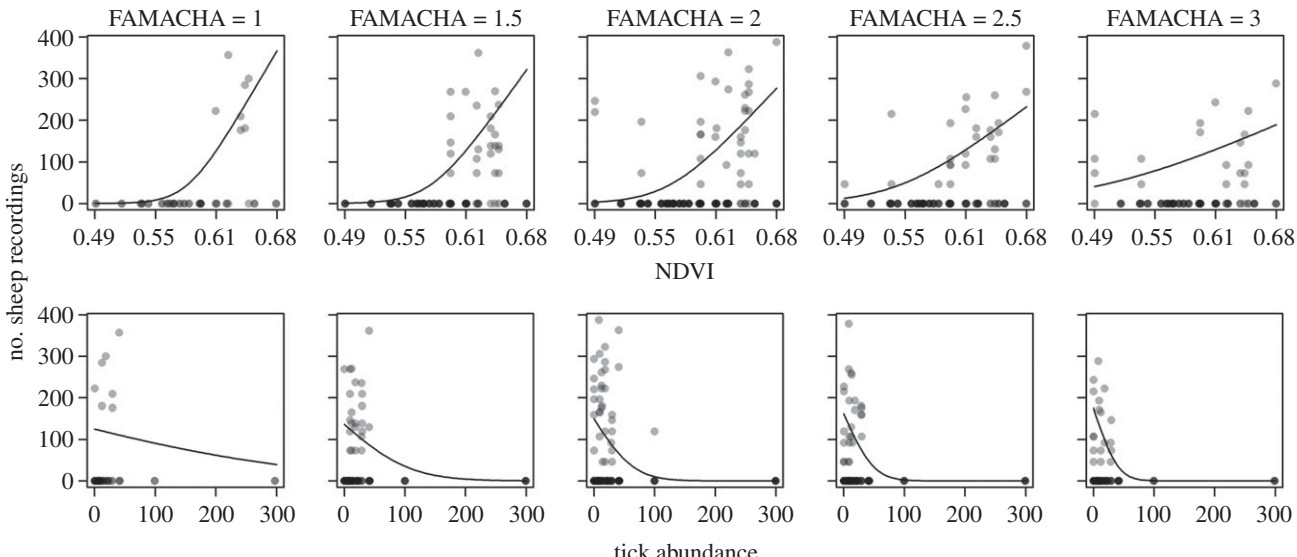

**Figure 2.** The effect of NDVI (top row) and tick abundance (bottom row) on the number of recordings of sheep with different FAMACHA scores in buffer zones of 50 m around the sampling points. FAMACHA scores range from 1 to 3, with 1 being the healthiest. The lines represent the fitted values calculated from the GLMM coefficients, while controlling for all other explanatory variables in the NDVI × FAMACHA (top row) and ticks × FAMACHA (bottom row) models at their mean value in the dataset. The points represent the actual number of sheep recordings in tick sampling sites, with different transparencies representing the number of sheep recordings (i.e. darker points equal more recordings). Figures for buffer zones of 30, 75 and 100 m are provided in electronic supplementary material.

## 4. Discussion

Ticks were found to be more abundant in areas with more productive vegetation (higher NDVI) [35]. Thus, grazers such as sheep potentially face a trade-off between maximizing forage intake and minimizing infection risk. The response of individual sheep to this trade-off varied with their anaemia status as measured by the FAMACHA score, even without very anaemic individuals (scores of 4 and 5) being recorded in the sample. While sheep in general were recorded more frequently around sampling points that had higher NDVI, the distribution of healthier, less anaemic sheep was more strongly associated with NDVI than less healthy individuals. Therefore, healthier individuals graze in more productive areas. This greater responsiveness of healthier individuals to vegetation productivity was reversed when considering tick abundance, as it was the less healthy individuals that showed a stronger (and negative) association with areas of high tick abundance. Less healthy individuals appeared to be avoiding areas with high tick abundances more than healthier individuals. This finding implies that healthy individuals prioritize forage intake while less healthy individuals prioritize parasite avoidance. The use of biologging in this study enabled us to quantify patterns of space use at the individual level and suggests a dependency on internal state which would not have been evident using overall sheep distributions alone.

Previously, sheep with high gastrointestinal parasite burdens have been found to avoid areas contaminated with faeces, in which parasite encounter risk is higher, to a greater extent than sheep with low parasite burdens [9]. This is likely to be due to the detrimental effect of these parasites, such as damage to gastrointestinal tissue, being greater in sheep which already carry a parasite burden, thereby increasing the cost of grazing in risky areas [36,37]. It has been hypothesized that the reduction in feeding motivation may allow a parasitized host to adopt a more selective feeding strategy,

thereby reducing any further ingestion of parasites and promoting an effective immune response [14,38]. Grazing by sheep in areas of high NDVI may lead to larger amounts of faeces and therefore higher infection risks in these areas, and this may drive use of lower NDVI areas by sheep with higher FAMACHA scores. Further sampling to include the distribution of faeces, and ideally experimental manipulation of this distribution, would help elucidate the cues sheep with differing health statuses use.

By contrast, evidence is mixed as to whether herbivores such as sheep can detect and avoid parasites not associated with cues as obvious as faeces, such as ticks. De Garine-Wichatitsky *et al*. [13] found that the ungulates they studied in Zimbabwe were unable to avoid encountering various species of *Rhipicephalus* ticks. Some species of this tick occurred in all vegetation types, thereby providing no indicators of their presence, while other species occurred in key forage resources such as in vegetation surrounding water holes which could not be avoided by hosts without incurring great nutritional costs. However, unlike the African *Rhipicephalus* ticks, common UK and European species of ticks such as *Ixodes ricinus* and *Dermacentor reticulatus* are associated with specific and thus predictable habitat types [29,39]. In general, parasites and their vectors are dependent on specific abiotic conditions with ticks preferring to inhabit dense matted vegetation in which humidity and warmth are sufficient and relatively stable [19,21]. Sheep may be able to avoid these specific habitat types to reduce encounter and hence infestation risk. Alternatively, sheep may avoid ticks directly as has been shown to be the case in white-tailed deer and some smaller mammalian hosts (squirrels and raccoons) shown to give up larger amounts of food in areas with a higher density of ticks [11,23]. Cattle have also been found to avoid grazing in paddocks with high densities of tick larvae [22] and actively avoided high concentrations of *Rhipicephalus microplus* tick larvae experimentally deposited

in patches onto pasture [39]. Past studies have been experimental, either controlling the abundance of ticks or the availability of food. Our study adds to past findings by suggesting avoidance of ticks distributed naturally.

In this study, less healthy individuals (based on FAMACHA scores) tended to be found more often in areas with lower tick abundances to a greater extent than healthier individuals, and showed a reduced response to NDVI, suggesting that for these individuals, the avoidance of areas with more ticks is more important than foraging in areas of more highly productive vegetation. Healthier sheep appeared to favour areas providing greater foraging intake despite the potential higher risk of infection from ticks. Studies have shown that sheep with higher crude protein diets are less affected by gastrointestinal nematode infections in terms of weight gain, anaemia and FECs, and vegetation with high NDVI has been shown to be of higher quality in terms of protein content [40–42]. Furthermore, hosts can acquire immunity, which increases with exposure to parasites and prevents subsequent infections, thereby reducing the cost of foraging in areas with high parasite abundances [43,44]. Our observational study cannot determine the behavioural and immunological mechanisms underlying space use by individuals but provides an important first step towards integrating internal and external factors in the study of spatial distributions of animals. Future studies could use controlled interventions, such as antiparasitic treatment, to separate cause and effect and develop understanding of the processes generating the observed associations. More extensive sampling of additional measures of individual health and parasite infection risk at each sampling site would further strengthen the study.

Understanding the spatial dynamics of habitat use by ungulates is critical for developing sustainable land management practices and biologgers are also increasingly being used in domestic livestock studies [45–48]. Our finding that the trade-off between foraging and parasite avoidance depends on the health status of individual sheep emphasizes the need to consider individual patterns in habitat utilization when trying to understand the distribution of grazing animals, since important factors and processes could be concealed in population-level studies. Biologgers provide a feasible means of obtaining such individual-level data, allowing individual behaviour to be considered alongside population-level patterns. This technology should be used more widely in future work to help tackle long-standing questions around parasite–foraging interactions.

Ethics. All procedures relating to sheep were approved by the University of Bristol Ethical Review Group (UIN UB/16/076).

Data accessibility. The datasets supporting this article have been uploaded as part of the electronic supplementary material.

Authors' contributions. All authors conceived the idea for this study. C.L. and K.B. designed the methodology and collected data. C.C.I. and C.L. analysed the data and wrote the manuscript. All authors contributed feedback to the drafts and gave final approval for publication.

Competing interests. We declare we have no competing interests.

Funding. This work was supported by an NERC GW4+ Doctoral Training Partnership studentship awarded to C.L. and an NERC Independent Research Fellowship (grant no. NE/K009370/1) awarded to C.C.I. E.R.M. is funded by BBSRC project no. BB/S014748/1.

Acknowledgements. We thank Helen Radmore for allowing us to work with her sheep, Richard Wall for support throughout the PhD project and Luca Börger for his feedback on an earlier version of this manuscript.

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
