## [Reviewer comments · Proceedings of the Royal Society B: Biological Sciences]

Review History

RSPB-2019-1581.R0 (Original submission)

Review form: Reviewer 1

Recommendation

Major revision is needed (please make suggestions in comments)

Scientific importance: Is the manuscript an original and important contribution to its field?
Marginal

General interest: Is the paper of sufficient general interest?
Acceptable

Quality of the paper: Is the overall quality of the paper suitable?
Acceptable

Is the length of the paper justified?
Yes

Should the paper be seen by a specialist statistical reviewer?
Yes

Do you have any concerns about statistical analyses in this paper? If so, please specify them explicitly in your report.

No

It is a condition of publication that authors make their supporting data, code and materials available - either as supplementary material or hosted in an external repository. Please rate, if applicable, the supporting data on the following criteria.

Is it accessible?

Yes

Is it clear?

Yes

Is it adequate?

Yes

Do you have any ethical concerns with this paper?

No

Comments to the Author

See attached. (See Appendix A)

Review form: Reviewer 2

Recommendation

Major revision is needed (please make suggestions in comments)

Scientific importance: Is the manuscript an original and important contribution to its field?

Good

General interest: Is the paper of sufficient general interest?

Excellent

Quality of the paper: Is the overall quality of the paper suitable?

Good

Is the length of the paper justified?

Yes

Should the paper be seen by a specialist statistical reviewer?

No

Do you have any concerns about statistical analyses in this paper? If so, please specify them explicitly in your report.

Yes

It is a condition of publication that authors make their supporting data, code and materials available - either as supplementary material or hosted in an external repository. Please rate, if applicable, the supporting data on the following criteria.

Is it accessible?

Yes

Is it clear?

Yes

Is it adequate?

Yes

Do you have any ethical concerns with this paper?

No

Comments to the Author

[formatted copy of my comments to author attached]

In general I liked this paper and very much appreciated the integration of a novel bio-logging approach with traditional field ecology to investigate a really interesting and novel research question. The paper is generally very well written (but see comments about results reporting, below). The results are tantalising and suggest that individual animals might make foraging / parasite avoidance trade-offs based on their own health status, but I have a few queries and comments about the way the data are handled and how the results are illustrated and reported that at the moment leave me a little uncertain about the validity of the findings. There are also a number of other comments that I would like the authors to address. It is good that the authors acknowledge (lines 288-290) that this is an observational study and so cause and effect are difficult to separate, and I agree with them that despite this the results are potentially of interest to a wider readership.

More significant concerns:

1. From the figures presented, the sheep observation data appear to be strongly zero-inflated (i.e. in most cases, sheep were not present), and yet there is nothing in the statistical analysis section to explain explicitly how this is dealt with by the authors. Such data types typically require particular approaches (hurdle models, etc). Can the authors ensure that the correct approach has been taken here, and give a more detailed explanation as to how this aspect is taken care of in the analysis?
2. Related to this, I am struggling to understand the figures 2a-c, particularly 2c. The variation between the fitted models for FAMCHA 1 thru' 3 appears to be driven solely by the datapoints for tick abundances between 0 and 50, and yet the green line (FAMCHA = 3, the 'least healthy' sheep) lies above all of the associated green datapoints. How can this be? This is the figure that should really be illustrating the key result, but this visual mismatch between fitted line and datapoints reduce confidence.
3. Another concern is that the measure of health status is indirect and one-dimensional, and although it is one that is used in the industry, there is a lack of fit between FAMACHA and FEC, which is unexpected. I would have expected a comment about the limitations of this measure in the discussion (or a better defence / justification for using FAMACHA).
4. In some places I found the explanations of the results to be too much focused on the statistical model output and too little on the biology / ecology of the system. For example, on lines 184-202 (but not only here), I find the explanation of these results very abstract. I understand that the authors are being careful to justify / explain their statistical models carefully, but in doing so the biological / ecological results have become obscured. In general, I would like to see some more clear explanations of what the results mean in the context of the study, alongside the explanation of the statistical models.

Minor comments, typos and suggested edits to clarify:

Line 43: could this string of citations be shortened to [8, 12-17]?

Line 43-47: Long convoluted sentence. Suggest splitting and altering the structure to read:

“Because the distribution of parasites in space and time influences their likelihood of being encountered, host behavioural defences will depend on the predictability of the parasites’ locations. This is particularly true because many parasites of terrestrial animals have lower mobility than their hosts, making their successful transmission highly dependent on the movement of the host rather than on the movement of the parasites themselves”

Line 52: Is the UK focus essential? Could you say ‘Northern Europe’ to increase relevance?

Line 58: Perhaps explain what ‘direct’ avoidance might look like?

Line 70: I miss a clear explanation of the major key knowledge gap / question of interest at the end of this paragraph. Essential to include this prior to the next ‘Aims’ paragraph.

Line 72: Latin binomial for sheep?

Line 75: Write out ‘normalized difference vegetation index’ in full first time, and give a brief explanation for the non-specialist.

Line 76: Suggest re-jigging the sentence to read: “The use of GPS allows the high resolution spatial and temporal tracking of individuals over extended periods of time”.

Line 88-89: Latin binomials, and use less technical terms for the non-agricultural reader

Line 135-140: This section describes the technical but not the ecological aspects of NDVI. More detail needed on what NDVI is, and what the satellite data means in terms of describing the forage type.

Line 165-166. The statement currently on lines 114-115 (‘Lower scores indicate less anaemia and better health [27]’) would actually be more useful and in context here.

Lines 166-167. Could these tick values be expressed as densities per m²?

Decision letter (RSPB-2019-1581.R0)

09-Sep-2019

Dear Ms Liddell:

I am writing to inform you that your manuscript RSPB-2019-1581 entitled "Response to resources and parasites depends on health status in extensively grazed sheep" has, in its current form, been rejected for publication in Proceedings B.

This action has been taken on the advice of referees, who have recommended that substantial revisions are necessary. With this in mind we would be happy to consider a resubmission, provided the comments of the referees are fully addressed. However please note that this is not a provisional acceptance.

1) A ‘response to referees’ document including details of how you have responded to the comments, and the adjustments you have made.

- 2) A clean copy of the manuscript and one with 'tracked changes' indicating your 'response to referees' comments document.
- 3) Line numbers in your main document.

Sincerely,
 Professor Hans Heesterbeek
 mailto: proceedingsb@royalsociety.org

Associate Editor

Comments to Author:

This is an interesting study using a novel approach that is of good general interest and quality. The two reviewers have identified several issues for consideration that they believe would result in an improved manuscript. In particular, one reviewer suggests that the conclusions should be toned down to take into account some of the design limitations, and the other reviewer has some constructive comments regarding the statistical treatment and reporting of key results.

Reviewer(s)' Comments to Author:

Referee: 1

Comments to the Author(s)
 See attached.

Referee: 2

Comments to the Author(s)
 [formatted copy of my comments to author attached]

In general I liked this paper and very much appreciated the integration of a novel bio-logging approach with traditional field ecology to investigate a really interesting and novel research question. The paper is generally very well written (but see comments about results reporting, below). The results are tantalising and suggest that individual animals might make foraging / parasite avoidance trade-offs based on their own health status, but I have a few queries and comments about the way the data are handled and how the results are illustrated and reported that at the moment leave me a little uncertain about the validity of the findings. There are also a number of other comments that I would like the authors to address. It is good that the authors acknowledge (lines 288-290) that this is an observational study and so cause and effect are difficult to separate, and I agree with them that despite this the results are potentially of interest to a wider readership.

More significant concerns:

1. From the figures presented, the sheep observation data appear to be strongly zero-inflated (i.e. in most cases, sheep were not present), and yet there is nothing in the statistical analysis section to explain explicitly how this is dealt with by the authors. Such data types typically require particular approaches (hurdle models, etc). Can the authors ensure that the correct approach has been taken here, and give a more detailed explanation as to how this aspect is taken care of in the analysis?

2. Related to this, I am struggling to understand the figures 2a-c, particularly 2c. The variation between the fitted models for FAMCHA 1 thru' 3 appears to be driven solely by the datapoints for tick abundances between 0 and 50, and yet the green line (FAMCHA = 3, the 'least healthy' sheep) lies above all of the associated green datapoints. How can this be? This is the figure that should really be illustrating the key result, but this visual mismatch between fitted line and datapoints reduce confidence.

3. Another concern is that the measure of health status is indirect and one-dimensional, and although it is one that is used in the industry, there is a lack of fit between FAMACHA and FEC, which is unexpected. I would have expected a comment about the limitations of this measure in the discussion (or a better defence / justification for using FAMACHA).

4. In some places I found the explanations of the results to be too much focused on the statistical model output and too little on the biology / ecology of the system. For example, on lines 184-202 (but not only here), I find the explanation of these results very abstract. I understand that the authors are being careful to justify / explain their statistical models carefully, but in doing so the biological / ecological results have become obscured. In general, I would like to see some more clear explanations of what the results mean in the context of the study, alongside the explanation of the statistical models.

Minor comments, typos and suggested edits to clarify:

Line 43: could this string of citations be shortened to [8, 12-17]?

Line 43-47: Long convoluted sentence. Suggest splitting and altering the structure to read: "Because the distribution of parasites in space and time influences their likelihood of being encountered, host behavioural defences will depend on the predictability of the parasites' locations. This is particularly true because many parasites of terrestrial animals have lower mobility than their hosts, making their successful transmission highly dependent on the movement of the host rather than on the movement of the parasites themselves"

Line 52: Is the UK focus essential? Could you say 'Northern Europe' to increase relevance?

Line 58: Perhaps explain what 'direct' avoidance might look like?

Line 70: I miss a clear explanation of the major key knowledge gap / question of interest at the end of this paragraph. Essential to include this prior to the next 'Aims' paragraph.

Line 72: Latin binomial for sheep?

Line 75: Write out 'normalized difference vegetation index' in full first time, and give a brief explanation for the non-specialist.

Line 76: Suggest re-jigging the sentence to read: "The use of GPS allows the high resolution spatial and temporal tracking of individuals over extended periods of time".

Line 88-89: Latin binomials, and use less technical terms for the non-agricultural reader

Line 135-140: This section describes the technical but not the ecological aspects of NDVI. More detail needed on what NDVI is, and what the satellite data means in terms of describing the forage type.

Line 165-166. The statement currently on lines 114-115 ('Lower scores indicate less anaemia and better health [27]') would actually be more useful and in context here.

Lines 166-167. Could these tick values be expressed as densities per m²?

Author's Response to Decision Letter for (RSPB-2019-1581.R0)

See Appendices B & C.

RSPB-2019-2905.R0

Review form: Reviewer 2

Recommendation

Accept as is

Scientific importance: Is the manuscript an original and important contribution to its field?

Good

General interest: Is the paper of sufficient general interest?

Excellent

Quality of the paper: Is the overall quality of the paper suitable?

Excellent

Is the length of the paper justified?

Yes

Should the paper be seen by a specialist statistical reviewer?

No

Do you have any concerns about statistical analyses in this paper? If so, please specify them explicitly in your report.

No

It is a condition of publication that authors make their supporting data, code and materials available - either as supplementary material or hosted in an external repository. Please rate, if applicable, the supporting data on the following criteria.

Is it accessible?

Yes

Is it clear?

Yes

Is it adequate?

Yes

Do you have any ethical concerns with this paper?

No

Comments to the Author

In general I liked this paper and very much appreciated the integration of a novel bio-logging approach with traditional field ecology to investigate a really interesting and novel research question. The paper is very well written. The results suggest that individual animals might make foraging / parasite avoidance trade-offs based on their own health status, and this is an important finding that would be of interest to a wide scientific audience.

I reviewed an earlier version of this manuscript and raised a number of queries around the statistical treatment and the presentation and description of the results. I am pleased to see that these have all been dealt with satisfactorily by the authors and as a result the paper is far tighter, more readable and the key messages are clearer. I commend the authors for their exemplary handling of responding to these queries.

Decision letter (RSPB-2019-2905.R0)

10-Jan-2020

Dear Ms Liddell

I am pleased to inform you that your Review manuscript RSPB-2019-2905 entitled "Response to resources and parasites depends on health status in extensively grazed sheep" has been accepted for publication in Proceedings B.

The referee and the Associate Editor do not recommend any further changes. Therefore, please proof-read your manuscript carefully and upload your final files for publication. Because the schedule for publication is very tight, it is a condition of publication that you submit the revised version of your manuscript within 7 days. If you do not think you will be able to meet this date please let me know immediately.

To upload your manuscript, log into <http://mc.manuscriptcentral.com/prsb> and enter your Author Centre, where you will find your manuscript title listed under "Manuscripts with Decisions." Under "Actions," click on "Create a Revision." Your manuscript number has been appended to denote a revision.

You will be unable to make your revisions on the originally submitted version of the manuscript. Instead, upload a new version through your Author Centre.

- 1) A text file of the manuscript (doc, txt, rtf or tex), including the references, tables (including captions) and figure captions. Please remove any tracked changes from the text before submission. PDF files are not an accepted format for the "Main Document".
- 2) A separate electronic file of each figure (tiff, EPS or print-quality PDF preferred). The format should be produced directly from original creation package, or original software format. Please note that PowerPoint files are not accepted.
- 3) Electronic supplementary material: this should be contained in a separate file from the main text and the file name should contain the author's name and journal name, e.g. `authorname_procb_ESM_figures.pdf`

All supplementary materials accompanying an accepted article will be treated as in their final form. They will be published alongside the paper on the journal website and posted on the online figshare repository. Files on figshare will be made available approximately one week before the accompanying article so that the supplementary material can be attributed a unique DOI. Please see: <https://royalsociety.org/journals/authors/author-guidelines/>

4) Data-Sharing and data citation

It is a condition of publication that data supporting your paper are made available. Data should be made available either in the electronic supplementary material or through an appropriate repository. Details of how to access data should be included in your paper. Please see <https://royalsociety.org/journals/ethics-policies/data-sharing-mining/> for more details.

If you wish to submit your data to Dryad (<http://datadryad.org/>) and have not already done so you can submit your data via this link <http://datadryad.org/submit?journalID=RSPB&manu=RSPB-2019-2905> which will take you to your unique entry in the Dryad repository.

Once again, thank you for submitting your manuscript to Proceedings B and I look forward to receiving your final version. If you have any questions at all, please do not hesitate to get in touch.

Sincerely,
Professor Hans Heesterbeek
<mailto:proceedingsb@royalsociety.org>

Associate Editor
Board Member
Comments to Author:

The authors have done a good job of revising their manuscript in response to the reviews - particularly in response to reviewer reservations about the statistical analysis and subsequent interpretation of the results - and the manuscript is now much improved as a consequence.

Reviewer(s)' Comments to Author:

Referee: 2

Comments to the Author(s).

In general I liked this paper and very much appreciated the integration of a novel bio-logging approach with traditional field ecology to investigate a really interesting and novel research question. The paper is very well written. The results suggest that individual animals might make foraging / parasite avoidance trade-offs based on their own health status, and this is an important finding that would be of interest to a wide scientific audience.

I reviewed an earlier version of this manuscript and raised a number of queries around the statistical treatment and the presentation and description of the results. I am pleased to see that these have all been dealt with satisfactorily by the authors and as a result the paper is far tighter, more readable and the key messages are clearer. I commend the authors for their exemplary handling of responding to these queries.

Decision letter (RSPB-2019-2905.R1)

14-Jan-2020

Dear Ms Liddell

I am pleased to inform you that your manuscript entitled "Response to resources and parasites depends on health status in extensively grazed sheep" has been accepted for publication in Proceedings B.

You can expect to receive a proof of your article from our Production office in due course, please

check your spam filter if you do not receive it. PLEASE NOTE: you will be given the exact page length of your paper which may be different from the estimation from Editorial and you may be asked to reduce your paper if it goes over the 10 page limit.

Your article has been estimated as being 8 pages long. Our Production Office will be able to confirm the exact length at proof stage.

Open Access

Paper charges

Sincerely,

Proceedings B

Appendix A

The paper reports on an interesting and useful study of sheep movement patterns in relation to environmental factors and measures of animal health. The lack of experimental interventions (e.g. use of drugs to control parasites) limits the ability of the (correlative) study to deliver strong general insights (see points below). For this reason, some of the conclusions should be toned down.

The paper centres around environmental distributions of ticks and forage resources (NDVI), amongst others, and the movement patterns of sheep with varying health status as measured by FAMACHA and FEC. The positive correlation between ticks and NDVI suggests the presence of a trade-off and the movement patterns of the sheep of varying health status are used to suggest the sheep perceive this trade-off. Whilst the results support the hypothesis, there are a number of assumptions made that limit the ability of this study to do more than make suggestions, including:

- (1) Health measures: FAMACHA is a measure of anaemia in response to *H. contortus* infection. No measures of tick burdens are reported. Are there data to support a correlation between tick infection rates and FAMACHA?
- (2) FEC is a concentration and varies greatly relative to the digestibility of the forage. Could this influence the results?
- (3) The ability of sheep to detect tick abundance is key to the interpretation of the results and the ability of the animals to detect the trade-off. Without experimental interventions, the supporting evidence for this is limited.
- (4) Why report environmental distributions of ticks but not animal burdens and animal measures of parasitism but not environmental burdens (e.g. levels of faecal contamination)?
- (5) No support for the use of NDVI as an indicator of forage quality at the study site?

For the above reasons statements like the one made in lines 246-248 should be toned down; 'reveals' should be 'suggests'. Similarly lines 276-277; 'showing' to 'suggesting'.

Appendix B

General comments

In general I liked this paper and very much appreciated the integration of a novel biologging approach with traditional field ecology to investigate a really interesting and novel research question. The paper is generally very well written (but see comments about results reporting, below). The results are tantalising and suggest that individual animals might make foraging / parasite avoidance trade-offs based on their own health status, but I have a few queries and comments about the way the data are handled and how the results are illustrated and reported that at the moment leave me a little uncertain about the validity of the findings. There are also a number of other comments that I would like the authors to address. It is good that the authors acknowledge (lines 288-290) that this is an observational study and so cause and effect are difficult to separate, and I agree with them that despite this the results are potentially of interest to a wider readership.

We would like to thank the reviewer for this encouraging feedback and are glad they feel that this is a novel research question of interest to a wide readership. We have done our best to address all the concerns and comments made, which we detail below.

More significant concerns:

1. From the figures presented, the sheep observation data appear to be strongly zeroinflated (i.e. in most cases, sheep were not present), and yet there is nothing in the statistical analysis section to explain explicitly how this is dealt with by the authors. Such data types typically require particular approaches (hurdle models, etc). Can the authors ensure that the correct approach has been taken here, and give a more detailed explanation as to how this aspect is taken care of in the analysis?

We agree with the reviewer that we should have considered zero-inflation due to the large number of zeros in the data set. Thus, we fitted negative binomial, zero-inflated negative binomial, poisson and zero-inflated poisson models to determine how the main effects (NDVI, steepness, ticks, FAMACHA and FEC) affect number of sheep recordings. We compared Akaike information criterion (AIC) values of the models and found the zero-inflated negative binomial to be the best fit. The statistical methods have been revised to explain why zero-inflated models were used (line 161-167), and the Results have been modified as required from using these models (line 205, 206, 210-211, 213-227). While the overall take-home messages of the original submission have not altered, we have now revised the manuscript to reflect the change in some of the results caused by switching to zero-inflated negative binomial models.

2. Related to this, I am struggling to understand the figures 2a-c, particularly 2c. The variation between the fitted models for FAMACHA 1 thru' 3 appears to be driven solely by the datapoints for tick abundances between 0 and 50, and yet the green line (FAMACHA = 3, the 'least healthy' sheep) lies above all of the associated green datapoints. How can this be? This is the figure that should really be illustrating the key result, but this visual mismatch between fitted line and datapoints reduce confidence.

We thank the reviewer for this comment and with hindsight agree the original figure, although key, did not show the trends clearly. We believe that the new version (figure 2) now shows the raw data and fitted trend lines (now based on the zero-inflated negative binomial models) clearly and the key result that NDVI is a stronger driver for healthier individuals and tick abundance a stronger driver for less healthy individuals.

3. Another concern is that the measure of health status is indirect and onedimensional, and although it is one that is used in the industry, there is a lack of fit between FAMACHA and FEC, which is unexpected. I would have expected a comment about the limitations of this measure in the discussion (or a better defence / justification for using FAMACHA).

This is a good point and we regret not making it clear in the original submission. FAMACHA is an indicator for anaemia, while the FEC provides an indicator for nematode infection. Possible causes of anaemia do include infection with nematodes such as Haemonchus contortus (barber's pole worm). In such cases where anaemia is caused by nematode infection, we would expect a positive correlation between FAMACHA scores and FEC, but anaemia can also be caused by other parasites and diseases, such as Fasciola spp. (liver fluke) [manuscript ref. 27]. Significant blood loss can also be caused by heavy tick infestations which are known to have a significant impact on the health of hill sheep in upland grazing habitats [manuscript ref. 21]. Thus, FAMACHA and FEC scores do not always correlate, especially if animals are infected with multiple parasites and/or the cause of anaemia is not primarily a nematode infection. Therefore, although FAMACHA appears to be a 'unidimensional' indicator of health status, it does integrate the consequences of several different parasite infections and is a fair measure of health, or at least as fair as any other measure. We have added further justification for using FAMACHA (line 118-121).

4. In some places I found the explanations of the results to be too much focused on the statistical model output and too little on the biology / ecology of the system. For example, on lines 184-202 (but not only here), I find the explanation of these results very abstract. I understand that the authors are being careful to justify / explain their statistical models carefully, but in doing so the biological / ecological results have become obscured. In general, I would like to see some more clear explanations of what the results mean in the context of the study, alongside the explanation of the statistical models.

This is a good point. We have revised the text (line 213-227) to focus more on the ecological meaning of the results, which we hope will improve the reader's experience. This includes restructuring this section of results, where previously we described the model results and then described the direction of the trends. We have now integrated these together so the direction (and ecological results) follow from the relevant model description.

Minor comments, typos and suggested edits to clarify:

Line 43: could this string of citations be shortened to [8, 12-17]?

Yes, this was an oversight. It is now corrected (line 43)

Line 43-47: Long convoluted sentence. Suggest splitting and altering the structure to read: "Because the distribution of parasites in space and time influences their **likelihood of being encountered, host behavioural defences** will depend on the predictability of the parasites' **locations. This is particularly true because** many parasites of terrestrial animals have lower mobility than their hosts, making their successful transmission highly dependent on the movement of the host rather than on the movement of the parasites themselves"

Thank you for the suggestion, we have made this change (line 43-47).

Line 52: Is the UK focus essential? Could you say 'Northern Europe' to increase relevance?

Northern Europe has been added to increase relevance (line 52).

Line 58: Perhaps explain what 'direct' avoidance might look like?

This sentence has been expanded to describe the avoidance more clearly (line 58-61).

Line 70: I miss a clear explanation of the major key knowledge gap / question of interest at the end of this paragraph. Essential to include this prior to the next 'Aims' paragraph.

The aim is now stated in the final sentence of this paragraph (line 72-73).

Line 72: Latin binomial for sheep?

Latin binomial Ovis aries was already mentioned early in the methods section (now line 91), together with breed of sheep used in this study. We have now added it to the abstract (line

12-13) and the introduction (line 75) as well. We have also added Latin binomials for cows and ponies described in methods section (line 92).

Line 75: Write out 'normalized difference vegetation index' in full first time, and give a brief explanation for the non-specialist.

We have made these changes (line 78 and 143-146).

Line 76: Suggest re-jigging the sentence to read: "The use of GPS allows the high resolution spatial and temporal tracking of individuals over extended periods of time".

We agree this reads better and have made the change (line 79-80).

Line 88-89: Latin binomials and use less technical terms for the non-agricultural reader.

We have added Ovis aries, Bos Taurus and Equus caballus for sheep, cows and ponies, respectively (line 91-92). We have also removed agricultural terms.

Line 135-140: This section describes the technical but not the ecological aspects of NDVI. More detail needed on what NDVI is, and what the satellite data means in terms of describing the forage type.

As described above, we have now added extra text describing NDVI as a measure of primary productivity with higher values indicating greener, more productive vegetation (line 143-146).

Line 165-166. The statement currently on lines 114-115 ('Lower scores indicate less anaemia and better health [27]') would actually be more useful and in context here.

We have moved this line as suggested (line 181).

Lines 166-167. Could these tick values be expressed as densities per m²?

We have included tick values expressed as densities per m² (line 183-184).

Appendix C

The paper reports on an interesting and useful study of sheep movement patterns in relation to environmental factors and measures of animal health. The lack of experimental interventions (e.g. use of drugs to control parasites) limits the ability of the (correlative) study to deliver strong general insights (see points below). For this reason, some of the conclusions should be toned down.

We are glad the reviewer sees the value in the study and thank the reviewer for their comments. We acknowledge the limitations of the study (307-309) and have added a sentence calling for such experimental interventions, which would be the next step in this research (line 276-278, 309-311).

The paper centres around environmental distributions of ticks and forage resources (NDVI), amongst others, and the movement patterns of sheep with varying health status as measured by FAMACHA and FEC. The positive correlation between ticks and NDVI suggests the presence of a trade-off and the movement patterns of the sheep of varying health status are used to suggest the sheep perceive this trade-off. Whilst the results support the hypothesis, there are a number of assumptions made that limit the ability of this study to do more than make suggestions, including:

- (1) Health measures: FAMACHA is a measure of anaemia in response to *H. contortus* infection. No measures of tick burdens are reported. Are there data to support a correlation between tick infection rates and FAMACHA?

*Heavy tick infestations are known to cause anaemia and have a significant impact on the health of hill sheep in upland grazing habitats (manuscript ref. 21). Therefore, there are strong grounds to expect heavy tick burdens to affect FAMACHA score. Moreover, we use FAMACHA also as a more general measure of health. While FAMACHA is used mainly to measure *Haemonchus* burden in sheep and goats, other possible causes of anaemia described by the developers of the FAMACHA method include other parasites and diseases such as infestation with *Fasciola* spp. (liver fluke). Indeed, fluke burden in UK sheep has been found to correlate with FAMACHA score (Olah et al. 2015). Tick burdens were not recorded due to various constraints, including time, the welfare consequences of prolonged handling of sheep, and unreliability of tick counts in heavily fleeced sheep. We have added extra lines on this in the methods section (line 118-121).*

*Olah S, van Wyk JA, Wall R, Morgan ER. 2015 FAMACHA©: A potential tool for targeted selective treatment of chronic fasciolosis in sheep. *Vet Parasitol* **212**, 188-92.*

- (2) FEC is a concentration and varies greatly relative to the digestibility of the forage. Could this influence the results?

We are assuming that the sheep all had similar diets and forage intake, due in part to the fairly simple composition of vegetation in the area (grass & bracken). FEC has been shown to vary mostly with faecal moisture content rather than diet, and faecal dry matter output is a function of body weight in grazing ruminants. Previous studies have found no relationship between diet and faecal nematode egg counts in goats (e.g. Ceriac et al. 2017). Faecal moisture varied little between individuals in the present study, as assessed by faecal consistency. Therefore, it is unlikely that diet was an important confounder of FEC.

*Cériac S, Jayles C, Arquet R, Feuillet D, Félicité Y, Archimède H, Bambou JC. 2017 The nutritional status affects the complete blood count of goats experimentally infected with *Haemonchus contortus*. *BMC Vet Res.* **13**, 326.*

- (3) The ability of sheep to detect tick abundance is key to the interpretation of the results and the ability of the animals to detect the trade-off. Without experimental interventions, the supporting evidence for this is limited.

We acknowledge the limitation of our observational study in the paper but feel our results still provide an important first step towards integrating internal and external factors in the study of spatial distributions of animals. We have expanded on the text discussing how individuals may detect tick abundance, either directly or indirectly. Other observational and experimental results show tick avoidance behaviour in animals [manuscript refs. 11, 22 and 23] and support our results and hypothesis. Admittedly further work is required to prove causation and mechanism, and we make this clear in a new sentence in the discussion (line 309).

- (4) Why report environmental distributions of ticks but not animal burdens and animal measures of parasitism but not environmental burdens (e.g. levels of faecal contamination)?

Our hypothesis concerns the relationship between animal health status (as affected by internal parasite burden and other factors), and spatial distribution of animals: specifically, whether internal state alters exposure to spatially aggregated parasites like ticks through its effects on behaviour. Therefore, the key measures are indeed correlates of internal parasite burden, and distribution of infective parasite stages in the environment (ticks). For completeness, it would have been interesting to also measure nematode stages in the environment and tick burdens on sheep: we attempted the former but found infective nematode larvae to be present at too low a density to provide usable data, while issues with tick sampling are listed above. We do agree with the reviewer that more extensive sampling of both individual health and of the parasite infection risk at each sampling site would strengthen the study, but this was not possible with the practical limitations. We have added text to the Discussion to raise and acknowledge this point (line 311-312).

- (5) No support for the use of NDVI as an indicator of forage quality at the study site?

There is extensive evidence that NDVI is an indicator of forage quality, including in upland grass and bracken habitats (manuscript ref. 1, 35, 40). We have expanded the paragraph on NDVI to include a better description of what this measure is (line 143-146)

For the above reasons statements like the one made in lines 246-248 should be toned down; 'reveals' should be 'suggests'. Similarly lines 276-277; 'showing' to 'suggesting'.
We have made these changes (line 265 and line 300).